# Facile Synthesis of Hollow Glass Microsphere Filled PDMS Foam Composites with Exceptional Lightweight, Mechanical Flexibility, and Thermal Insulating Property

**DOI:** 10.3390/molecules28062614

**Published:** 2023-03-13

**Authors:** Tian-Long Han, Bi-Fan Guo, Guo-Dong Zhang, Long-Cheng Tang

**Affiliations:** 1Hangzhou Vocational & Technical College, Hangzhou 310018, China; 2Key Laboratory of Organosilicon Chemistry and Material Technology of MoE, College of Material, Chemistry and Chemical Engineering, Hangzhou Normal University, Hangzhou 311121, China

**Keywords:** polydimethylsiloxane foam, hollow glass microsphere, silicone chemistry, thermal insulation, lightweight

## Abstract

The feature of low-density and thermal insulation properties of polydimethylsiloxane (PDMS) foam is one of the important challenges of the silicone industry seeking to make these products more competitive compared to traditional polymer foams. Herein, we report a green, simple, and low-cost strategy for synthesizing ultra-low-density porous silicone composite materials via Si-H cross-linking and foaming chemistry, and the sialylation-modified hollow glass microspheres (m-HM) were used to promote the HM/PDMS compatibility. Typically, the presence of 7.5 wt% m-HM decreases the density of pure foam from 135 mg/cm^−3^ to 104 mg/cm^−3^ without affecting the foaming reaction between Si-H and Si-OH and produces a stable porous structure. The optimized m-HM-modified PDMS foam composites showed excellent mechanical flexibility (unchanged maximum stress values at a strain of 70% after 100 compressive cycles) and good thermal insulation (from 150.0 °C to 52.1 °C for the sample with ~20 mm thickness). Our results suggest that the use of hollow microparticles is an effective strategy for fabricating lightweight, mechanically flexible, and thermal insulation PDMS foam composite materials for many potential applications.

## 1. Introduction

Along with the enormous requirement of the buildings and packaging areas, the development of thermal insulation materials has become more critical. The building environment accounts for a substantial part of the global energy use, with 30% of the global CO_2_ emissions coming from the building industry. Building heating and cooling account for more than 10% of the global energy consumption [1]. Therefore, as a means to obtain the low thermal conductivity material, extensive research has reported aerogel materials (e.g., silica aerogels and nanocellulose aerogels) and polymer foam (e.g., polystyrene and polyurethane (PU)) [2,3,4]. While polymer foams have played a key role as a commercially available and highly workable insulating material, their vulnerability to high-temperature properties limits their applicability. Unfortunately, most aerogels have poor mechanical properties and high costs, though they exhibit very low density and thermal conductivity. Comparatively, polydimethylsiloxane (PDMS) foam materials are gaining considerable interest in both academic and industrial fields due to their excellent mechanical flexibility, thermal insulation properties, low density, and unique porous structures, and among them, they have rationalized the growth of the silicone product market [5,6,7], showing an ideal alternative for multifunctional emerging applications.

As is well known, PDMS foam materials usually show excellent thermal stability (from −80 °C to 240 °C) [8,9,10], good mechanical flexibility, and excellent chemical resistance [11], which is attributed to the high dissociation energy and low-energy barriers to Si-O-Si bond rotations [12,13,14]. Therefore, PDMS foam and its composites have been widely developed for versatile applications, such as wearable electronics [15,16], piezoresistive sensors [16], water/oil separation and absorbents [17], and thermal insulation materials [18]. It should be noted that the density of commercial PDMS foam material is usually >200 mg/cm^−3^, which is more than those of the traditional hydrocarbon polymer foam materials, e.g., 50–150 mg/cm^−3^ polyurethane or polystyrene foam, greatly limiting their application in the automobile field, although their other properties are superior to other polymer foams [19,20,21,22,23,24,25]. At present, PDMS foam production with a relatively high density still remains a challenge, which would be addressed by optimizing the material’s composition design and the fabricating process.

For the thermal insulating mechanism, the thermal conductivity and heat transport of thermally insulating materials are commonly described as a summation of contributions from natural convection, radiation, solid conduction, and gas conduction [1,26]. Normally, the natural convection would be restricted for the porous foam materials with microscale pore structures, and the thermal radiation is mainly determined by the temperature range. Hence, the solid and gas conductions are influenced by using porous foam materials. Generally, reducing the density of the porous materials would mitigate the affection of heat transfer by solid conduction [27], thereby improving the foam’s thermal insulation capabilities. Nevertheless, the contribution of gas conduction or gas convection to heat transport will increase due to the increased amount of gas. Some researchers have shown that radiative heat transfer needs to be accounted for when the density of a porous material is low [28]. Thus, preparing a prominent thermal insulating PDMS foam requires carefully balancing the trade-off relationship between solid thermal conduction and gas conduction in heat transfer.

To resolve the above issues, various strategies, including NaCl and the sugar template-based method [18,29,30,31,32], solvent–evaporation-assisted strategy [33,34,35], 3D printing technique [36,37,38], and chemical foaming [39,40], have been developed to reduce the density and, thus, improve the thermal insulation performance of the PDMS foam materials. For example, Abshirini et al. used a hybrid Toluene/THF mixture to fabricate porous PDMS structures with tunable microstructures by using a two-step phase separation method [41]. However, the fabricating processes of the as-prepared PDMS foam sample involve in-length and complicated postprocessing, and its density is much higher than other types of polymer foams. On the other hand, the PDMS foams prepared by choosing 3D interconnected networks of few-layer hexagonal boron nitride foam as supporting frameworks have demonstrated producing a low density of 15 mg cm^−3^, but the complex processing method cannot be directly used for large-scale production [42]. Therefore, it is still a challenge to develop a simple, green, and efficient strategy while avoiding more energy and time spent on the postprocessing to produce lightweight, mechanically flexible, and thermally insulating PDMS foam materials.

Herein, we report reproducible PDMS foam composite materials with the low density of ~100 mg/cm^−3^ and good mechanical properties by introducing silane-modified hollow glass microspheres (m-HM) via a simple mechanical mixing and chemical foaming strategy. Typically, the presence of m-HM particles can be well dispersed in the PDMS matrix via using a chemical modification of n-propyltrimethoxysilane. The effect of the content of m-HM on the morphology and porous structure of the PDMS foam materials was observed and analyzed. The density, mechanical flexibility and cyclic compressive reliability of the PDMS foam composites with different contents of m-HM were also compared and discussed. Finally, the thermal insulation performance of various m-HN-filled PDMS foam composites were investigated and compared. Overall, the design and development of introducing m-HM in phenyl silicone rubber foam materials in this work could provide a novel platform for next-generation polymer foam materials.

## 2. Results and Discussion

Figure 1 shows the schematic illustration of the structure formation of the silicone foam composites with different m-HM particles (abbreviated as SiRF-m-x, where x indicates the content of the m-HM particle). As shown in Figure 1a, the m-HM and PDMS molecules (i.e., PDMS-H and PDMS-Ph-OH) were mechanically mixed to obtain highly dispersed m-HM/PDMS suspension (Figure 1a). Then, the PDMS mixture (PDMS-Ph-Vi and PDMS-Ph-OH) and catalyst were added into the above suspension to prepare the SiRF@m-HM composite foam materials (Figure 1(ai) after the foaming reaction was conducted via releasing hydrogen gases from dehydrogenation reaction between Si-OH of PDMS-Ph-OH or H_2_O and Si-H of PDMS-H (Figure 1(bi,iii) [43,44,45]. Meanwhile, the mixture started cross-linking reaction by a typical hydrogenation reaction under Pt catalyst at ambient temperature (Figure 1(bii)) [46,47,48]. Finally, the m-HM/PDMS foam composites were successfully manufactured (Figure 1(aii)) within several minutes. It should be noted, during the foaming and cross-linking process, the mHGM particles would be kept inside the foam skeleton and some of them would be self-assembled onto the foam surface zone.

In order to prepare low-density and thermal insulation SiRF materials, commercial hollow glass microspheres were selected to be low-density fillers in SiRF composites. Typically, the modified HM particles are spherical (5–100 μm in diameter) with a smooth surface and an average thickness of ~310 nm wall, as observed by the SEM (Figure 2a,b), implying the shape uniformity and sphericity feature. Notably, the microscale hollow glass particles with the thin thickness implies a relatively large size of the hollow structure, which would play an important role in influencing the thermal insulating performance of the foam material (discussed later). Typically, the average diameter of the hollow microsphere is counted to be about 32.46 μm via Nano Measurer software, as shown in Figure 2c. Moreover, the HM particles are mainly composted of inorganic compound and cannot affect the high temperature tolerance of the PDMS foam materials. The element compositions of the HM particles were conducted by SEM-EDS mapping and the typical element compositions are shown in Figure 2d. Clearly, the smooth surface of the hollow glass microsphere is B of 2%, O of 51%, Na of 6%, Si of 39%, and Ca of 3%, respectively, suggesting a large amount of SiO_2_ composition are dominant in the HM particles.

As shown in Figure 3a, the phase separation phenomenon, of precipitating HM from silicone oil, can be observed after directly mixing the HM and PDMS mixture after two days, which would be attributed to incompatibility between the organic group of PMDS and hollow glass microspheres with Si-OH [11]. As a result, it is obvious that abundant centimeter-level pores are generated in the SiRF-5 material (Figure 3c), and the curing time of the SiRF-5 experimental group needs more time than 0.5 h. Within our research process, the cross-linking reaction is inhibited to form a foam when the added amounts of hollow glass microspheres are more than 5 wt% due to a structured phenomenon caused by the Si-OH of HM and the large specific surface area of HM. To address the above issue, the Si-OH of HM particles was chemically modified using n-propyltrimethoxysilane as a capping agent [11], preparing the modified hollow microspheres with a certain of hydrophobicity.

After replacing m-HM with untreated HM particles, the m-HM/PDMS mixture can remain stable even after 360 days without phase separation and present as highly viscous (Figure 3b), indicating that propyltrimethoxysilane molecules were successfully grafted to the surface of the hollow glass microspheres, which would promote the compatibility between the HM particles and the PDMS prepolymer molecules. At the same time, the SiRF-m-5 sample shows a delicate surface and uniform pore structure (Figure 3d) after about 7 min of the curing reaction due to the replacement of the silicone hydroxyl group of the m-HMs by organic groups. As shown in Figure 3e, a smooth pore surface and micron-level pore are found in SiRF-m-5. Moreover, the m-HM particles with a size of about 22 μm are well dispersed in their skeleton of silicone foam without a pore surface (Figure 3f), furthermore confirming the sialylation modification is beneficial for promoting the compatibility between the HM particle and the PDMS prepolymer.

During the synthesis of SiRF@m-HM composites, the amounts of sialylation reagent are fixed at 5 wt%, while the m-HM contents are in the range of 2.5–30 wt%. As shown in Figure 4, different sample morphologies can be obtained by introducing the m-HM content. Digital photos clearly indicate that a strong relationship between foam density and the amount of m-HM exists. In other words, when the m-HM loading is less than 7.5 wt%, the higher m-HM content produces lower density of the PDMS foam composites. Moreover, the foam pore size and foaming rate (final foam volume: initial mixed solution volume) increase in line with the increased m-HM content (Figure 4a–f), and the foam network structure is damaged gradually. It is hypothesized that a higher amount of m-HM in the PDMS phase could inhibit the curing reaction rate between Si-H of PDMS-H and Si-vi of PDMS-Ph-vi [49]. As in previous studies, the most significant factor was the balance of the reaction rates of foaming and cross-linking [50,51]. Due to the slower curing process, hydrogen gas from the dehydrogenation reaction between Si-OH of PDMS-Ph-OH or H_2_O and Si-H of PDMS-H converges into more giant bubbles or escapes directly into the air, which leads subsequently to obtaining the SiRF foam composites with an irregular network structure and a relatively high density (Figure 4e,f). Consequently, the foaming ratio and density of the SiRF foam composites decrease with the increased m-HM content when the m-HM loading is more than 15 wt%. As the m-HM content in the phenyl silicone foam increases, the pore size of the SiRF@m-HM composite gradually increases, resulting in the appearance of a discontinuous foam skeleton. Thus, the SiRF with high filler exhibits a more brittle structure compared to a SiRF composite with low-content m-HM. Eventually, the SiRF-m-7.5 composite possesses the best foaming ratio and interconnected cells, which is typical for close-cell foams. It is considered that the introduction of m-HM allows for achieving the low-density SiRF composites with well-defined porous morphologies.

The thermal insulation performance is a crucial index for practical applications of silicone foam as thermal insulating materials. Considering the gradient distribution of the thermal insulating materials on a high temperature surface, the direct obversion of temperature difference for both the hot and cold surfaces of the porous foam materials is effective for understanding the thermal insulation performance. To visually assess the thermal insulation properties of various SiRF composites, a setup was intentionally designed to detect the top face temperature (TFT) of the foam samples at an 80 °C hot stage by using an infrared thermal image tester. As shown in Figure 5a, silicone foam materials exhibit particularly favorable thermal insulation properties, owing to their porous structure and low thermal conductivity [18,22,52]. The thermal insulation properties of our prepared m-HM-filled SiRF composites were measured and shown in Figure 5b–f.

The results presented in Figure 5 show the top face temperature of the various SiRF composites at the hot stage of 150 °C by using a thermal IR image tester, and the thickness values of all samples were kept as ~10 mm. Typically, the TFT values of all samples exhibit an increase with increasing the time due to the thermal conduction effect. As expected, pure SiRF material shows excellent thermal insulation properties, with an average TFT of 54 °C after 1800 s, and the average TFT of pure SiRF is stable within 300 s on the 150 °C hot platform. With the increase of m-HM loading in the SiRF, the average TFT value of the SiRF@m-HM composites with an appreciate content of 2.5–7.5 wt% at 1800 s gradually reduces from 54 °C to 52.1 °C after 1800 s, as shown in Figure 5a–c. This suggests the improved thermal insulating performance after addition of the m-HM. The SiRF composites filled with 7.5 wt% m-HM display the lowest TFT of 52.1 °C (Figure 5c), which is well consistent with the observed changes of the pore structure shown in Figure 3. In other words, the thermal insulation performance of SiRF is positively correlated with the density when the m-HM loading is lower than 7.5 wt%. At the same thickness, introducing more modified hollow glass microspheres in the SiRF is beneficial to increase the interfacial thermal resistance and reduce the contribution of gas thermal conductivity in heat transfer [1]. For the SiRF composites filled with high contents of m-HM particles from 15.0 to 30 wt%, the TFT of the m-HM/SiRF composites increased sharply from 56.9 °C to 63.4 °C, and their average TFT stabilized after 900 s. The average TFT of SiRF-m-15 (52.2 °C) is higher than the average TFT of SiRF-m-2.5 (56.9 °C) after 150 °C heat processes for 30 min. Since the m-HM/SiRF has millimeter-sized pores, as well as open-cell pores, the contribution of heat convection to thermal radiation is enormous and cannot be ignored. Therefore, it is important to prepare the PDMS foam composite product by carefully optimizing the fillers loading inside the PDMS skeleton without affecting the formation of the good porous structure. The above results indicate that the presence of m-HM in SiRF composite materials improves the thermal insulation effectively at a certain filler content.

Based on the above experimental results, the TFT values of the SiRF@m-HM composites containing various filler contents as a function of time were summarized and shown in Figure 6a. The density values of the above SiRF@m-HM composites were also measured and shown in Figure 6b. Clearly, the thermal insulating performance of pure SiRF and SiRF@m-HM composites is strongly related with their density change. Considering the almost unchanged pore size shown in Figure 4, the presence of low contents (2.5–7.5 wt%) of m-HM particles can improve the TFT values, which is likely attributed to the thermal insulating effect of the hollow structure of m-HM particles in the foam skeleton. With increasing the content of the m-HM particle, the pore structure displays obvious change from close cell to open cell structure, and this would induce the decreased thermal condition of gas and skeleton. As a result, although SiRF-m-2.5 and SiRF-m-15 have similar densities (~113 mg/cm^−3^), the thermal insulating performance shows obvious differences. The addition of high contents of m-HM particles not only increases the density value from 113 mg/cm^−3^ to 266 mg/cm^−3^ when the m-HM loading is more than 15 wt% (Figure 5d–f) but also induces a much larger pore cell size (Figure 4), thus leading to the above different thermal insulating performances.

Generally, a desirable polymer foam materials should be mechanically robust and have a stable compressive stability performance for practical application, e.g., as seat cushion material in new energy vehicles. To evaluate the mechanical robustness and flexibility, the compressive strain–stress curves of the prepared m-HM-filled SiRF composites with an optimized content of 7.5 wt% were conducted, and the results are shown in Figure 6c–e. Normally, the stress–strain curves of the SiRF-m-7.5 sample under different strain values of 10, 20, 40, 60, and 80% in Figure 6c are stable and consistent, and the applied strain and size could recover the original stress and size values well once the applied stress was removed, revealing excellent and stable mechanical flexibility (see the inset in Figure 6c). Meanwhile, the SiRF-m-7.5 composite sample also exhibits good structure stability, and it can maintain the original shape and size even after 100 compressive cycles at R.T. (applying 70% strain), as shown in Figure 6d. This demonstrates the good mechanical robustness and flexibility of the porous structure of the SiRF composites filled with the optimized content of m-HM. The maximum stress value at 70% strain after 100 cycles is almost unchanged (see Figure 6e). Based on this result, we compared the density, processing condition, mechanical properties, and thermally insulating properties of various PDMS foam systems, which are given in Table 1. It can be seen that our foam possesses not only a simple fabrication process and mechanical flexibility but also exhibits ultra-low-density and excellent thermally insulating properties. These indicate that our simple and effective strategy provides a novel design and development of lightweight, mechanically flexible, and efficiently thermally insulating PDMS foam composites.

## 3. Experimental Section

### 3.1. Materials

Divinyl polymethylphenylsiloxane (PDPS-Vi) and dihydroxy polymethylphenylsiloxane (PDPS-OH) were obtained from Guangzhou Leso Chemical Technology Co., Ltd. (Guangzhou, China) Poly(methylhydrosiloxane) (PDMS-H) was purchased from Zhejiang Xinan Chemical Industrial Group Co., Ltd. (Hangzhou, China). Karstedt’s catalyst-diluted Pt was purchased from Betely Polymer Materials Co., Ltd. (Guangzhou, China). The hollow glass microsphere was obtained from China Steel Group Maanshan Mining Institute New Material Technology Co., Ltd. (Maanshan, China). Inhibitors, n-propyltrimethoxysilane, and ethanol were obtained by our laboratory and Sinopharm Chemical Reagent Co., Ltd. (Shanghai, China), respectively. In our work, all chemical reagents needed no purification further and were used as received.

### 3.2. Fabrication of PDMS Foam Composites Containing m-HM Particles

The fabricating process of PDMS foam composites containing m-HM particles was introduced as follows. First, the hollow glass microsphere was dispersed in water/ethanol (*v*/*v* = 1/9) and added to the three-necked bottles. Then, the pH mixture solution was adjusted to 9 and heated at 80 °C, and n-propyltrimethoxysilane was added to the above mixture at 80 °C for 5 h, followed by cooling naturally at room temperature (R.T.), obtaining a uniform and stable m-HM-based mixture. The mixture was centrifuged at 5000 rpm for 10 min, collecting a white sediment. Finally, the sediment was dried in an oven at 80 °C for 24 h, obtaining a white powder (m-HM).

The silicone rubber foam (SiRF), SiRF@HM, and SiRF@m-HM composites were fabricated by using a simple mechanical mixing approach, and the fabrication process was expressed as follows. First, the PDMS pre-polymer mixture was prepared by mechanically mixing the PDMS-H, PDMS-Ph-OH, inhibitor, water, and a certain amount of HM (or m-HM) at a speed of ~1100 rpm for 7 min to obtain a uniform mixture. Then, the PDMS mixture and Pt catalyst were added into the above mixture. The screw speed was 1200 rpm, and the time was set to 30 s. Subsequently, the mixture was poured into molds and foamed for 20 min at room temperature. Finally, the sample was placed into an oven at 80 °C for 5 h to obtain a PDMS/m-HM foam composite.

### 3.3. Characterization

The morphology and microstructure of various m-HM-filled PDMS foam composites were analyzed by scanning electron microscopy (SEM) (Sigma-500, ZEISS, Germany) with an energy dispersive spectrometer (EDS). The compression tests of foams (10 × 10 × 10 mm) were measured by using a universal tensile testing machine (UTTM) (Universal V4.5A models, China). The infrared thermography of the samples (10 × 10 × 10 mm), while conducting vertical burning tests, were obtained by an infrared thermal camera (FLIR) (Nicolet 7000, American) with a thermal sensitivity of ≤2 °C. The thermal insulation of the PDMS foam composites (thickness of ~20 mm) was conducted at the fixed hot stage with 150 °C, and the infrared images of the samples during the thermal insulating process were recorded using an infrared camera (FLIR E60) with a certain range of detection from −10 to 650 °C.

## 4. Conclusions

In summary, we developed a simple and green method for preparing lightweight, mechanically flexible, and thermal insulating phenyl silicone foam composites. Incorporation of m-HM particles into the PDMS prepolymer produced a good compatibility of the HM and PDMS mixture, thus fabricating the porous m-HM-filled PDMS composites with an excellent porous pore structure. The pore structure and density of the PDMS foam composites are strongly dependent on the content of the m-HM particles. After optimizing the filler content, the SiRF-m-7.5 composites display a density as low as 104 mg/cm^3^ and good thermal insulation performance (e.g., from 150.0 °C to 52.1 °C for the sample with ~20 mm thickness), which is superior to other SiRF composites with different m-HM contents. Moreover, the optimized PDMS foam composites also have excellent mechanical flexibility and reliability—for instance, unchanged maximum stress values at a strain of 70% after 100 compressive cycles. Clearly, this green, simple, and low-cost method was proven herein to be highly reproducible and capable of generating ultra-low-density PDMS foam composites for potential large-scale applications.

## Figures and Tables

**Figure 1 molecules-28-02614-f001:**
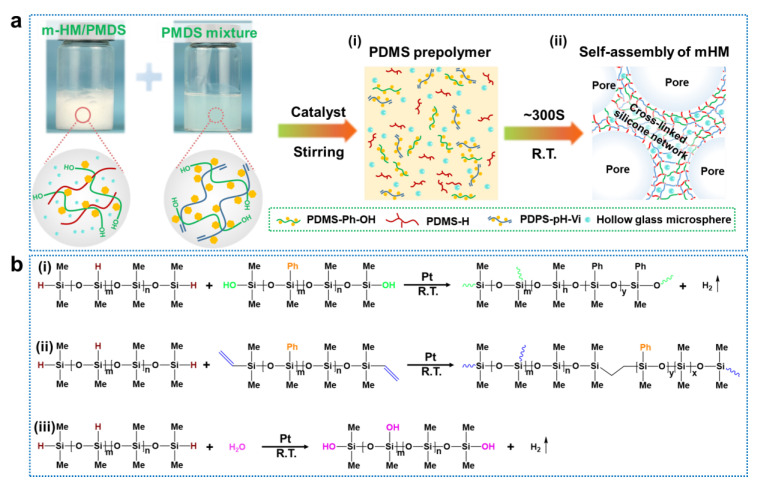
Design and reaction mechanism of PDMS foam composites filled with hollow glass microspheres: (**a**) Schematic processes of the fabricating process of hollow glass microspheres on the SiRF surface: (**i**) PDMS prepolymer and (**ii**) the dispersion of hollow glass microspheres in the foam skeleton. (**b**) Schematic illustration of cross-linking and foaming reactions used in synthesis of m-HM-filled SiRF nanocomposites: (**i**,**iii**) foaming reaction and (**ii**) cross-linking reaction.

**Figure 2 molecules-28-02614-f002:**
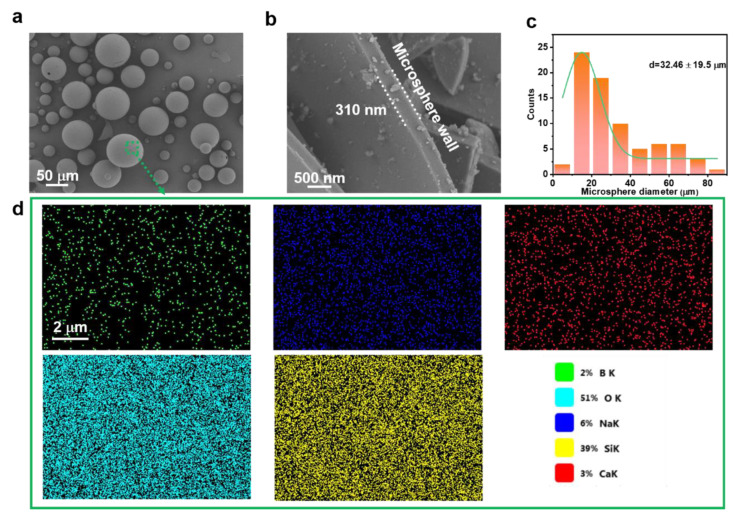
Structure and morphology of hollow glass microsphere: (**a**,**b**) SEM image of the planar and cross-sectional views of modified hollow glass microspheres. (**c**) The diameter size distribution of hollow glass microspheres, and (**d**) EDS mapping images of hollow glass microspheres.

**Figure 3 molecules-28-02614-f003:**
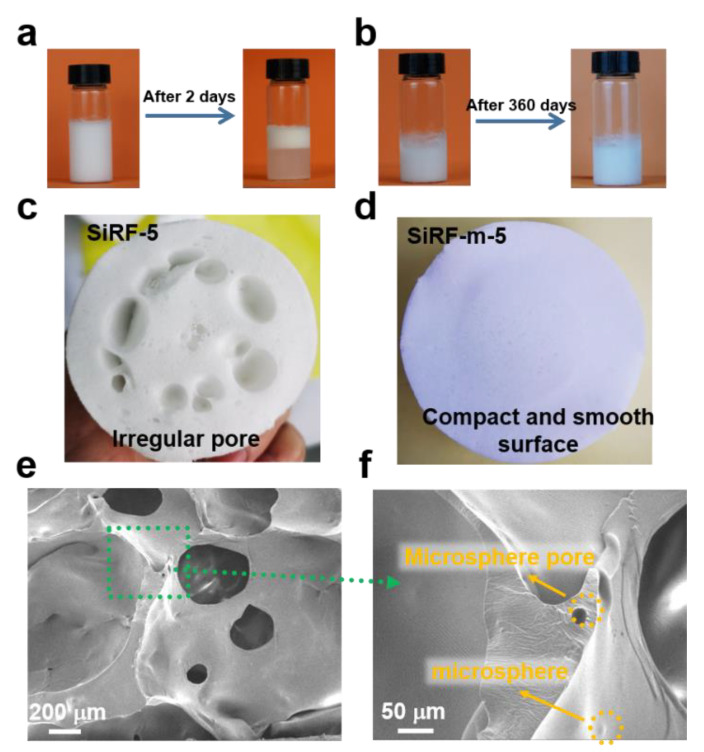
Sample fabrication and cell structure: (**a**,**b**) Digital photos of the HM-filled PDMS mixture and the m-HM filled PDMS mixture. (**c**,**d**) Digital photos of SiRF@HM and SiRF@m-HM. (**e**,**f**) SEM image of PDMS foam composites with 5.0 wt% m-HM particles (SiRF-m-5).

**Figure 4 molecules-28-02614-f004:**
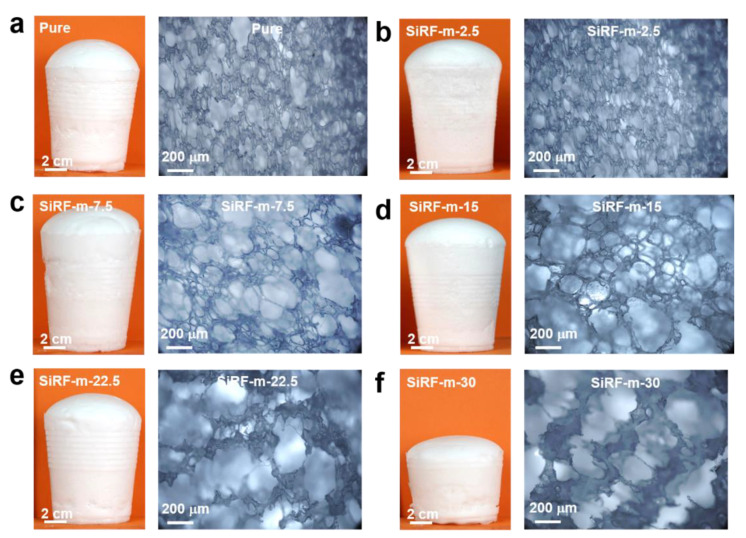
Structure and morphology observations: Digital photos and optical microscope images of SiRF composites with different m-HM contents: (**a**) pure SiRF, (**b**) SiRF-m-2.5, (**c**) SiRF-m-7.5, (**d**) SiRF-m-15, (**e**) SiRF-m-22.5, and (**f**) SiRF-m-30, respectively.

**Figure 5 molecules-28-02614-f005:**
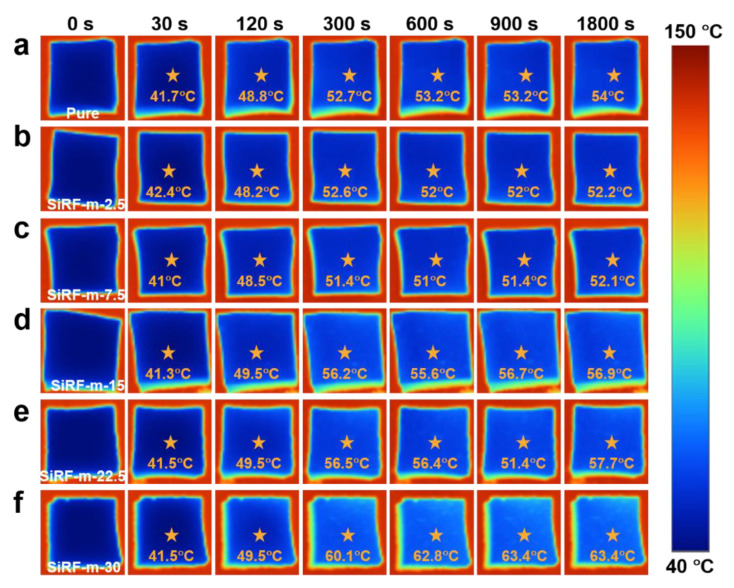
Thermal insulating performance and comparison: The infrared thermography of SiRF composites with different times at the hot stage of 150 °C: (**a**) pure SiRF, (**b**) SiRF-m-2.5, (**c**) SiRF-m-7.5, (**d**) SiRF-m-15, (**e**) SiRF-m-22.5, and (**f**) SiRF-m-30.

**Figure 6 molecules-28-02614-f006:**
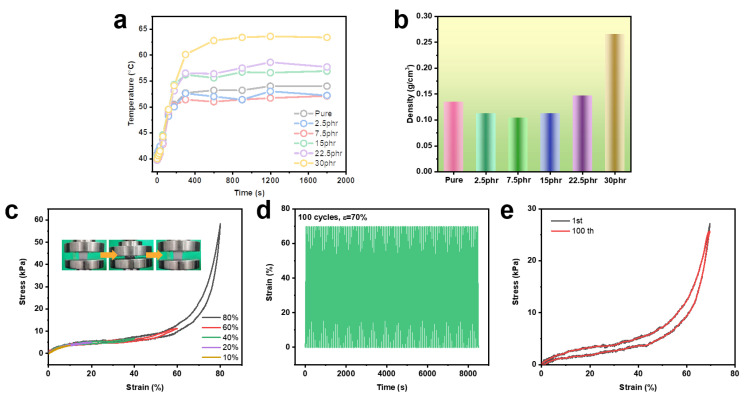
Thermal insulation, density, and mechanical properties of SiRF@m-HM composites: (**a**) The temperature–time curves of various SiRF composites as a function of time. (**b**) The density values of SiRF composites with different m-HM contents. (**c**) Compressive stress–strain curves of SiRF-m-7.5 with different strain values. (**d**) Cyclic compressive stress–time curves of SiRF-m-7.5 composites for 100 compressive cycles. (**e**) Typical compressive stress–strain curves of SiRF-m-7.5 composites under the 1st and 100th cycle.

**Table 1 molecules-28-02614-t001:** Comparisons of the processing condition, density, mechanical properties (εmax and stress), and thermally insulating properties (T1/T2/thickness).

Type ^a^	Processing Condition	Density (g/cm^−3^)	ε_max_ (%)/ Stress (kPa) ^c^	T_1_ (°C)/T_2_ (°C)/ Thickness (mm) ^d^	Ref.
Graphene/PDMS	Freeze-drying, annealing at 200 °C	74	~90%/~1845	61.2/200/-	[53]
ST-8-0.6	3D printing, drying at 80 °C	320	~60%/~400	NM	[36]
S46-40	Mechanical mixing, curing at 200 °C	~800	-/~11,000	NM	[14]
PDMS-40%/RFSi-0.6	“Co-gel” technique, drying at 80 °C	NM ^b^	20%/699	~175/-/4	[52]
BNF@PDMS	Ni template, curing at R.T.	15	~80/~1.5	NM	[42]
Sample-11	Mechanical mixing, foaming at 70 °C	346	~70%/~520	NM	[35]
SiRF-m-7.5	Mechanical mixing, foaming at R.T.	104	80%/58	52.1/150/20	This work

^a^ ST: simple tetragonal; S46-40: 40 wt% hollow glass micro-balloons of S46; RF: resorcinol formaldehyde; h-BNFs: hexagonal boron nitride foams; Sample-11: PDMS base/Span 80. ^b^ NM: Not Mentioned. ^c^ ε_max_: Maximum strain. ^d^ T_1_: top temperature of objective; T_2_: test temperature.

## Data Availability

Not applicable.

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
