# Peer review of "Facile Synthesis of Hollow Glass Microsphere Filled PDMS Foam Composites with Exceptional Lightweight, Mechanical Flexibility, and Thermal Insulating Property"

_molecules, 2023, doi:10.3390/molecules28062614_

Round 1

Reviewer 1 Report

In this manuscript, the authors perform surface modification (m-HM) of insulating glass microspheres by adding a silane. In this way, the compoundability of HM/PDMS is improved, and polydimethylsiloxane (PDMS) can be formed into porous silicone composites. According to the authors, porous silicone resin composites not only have low density, but also have good properties in material thermal insulation and mechanical toughness. Thus, I recommend publication in Molecules after the authors address the following minor comments.

1. In the manuscript, with the increase of m-Hm content, there is not enough data and pictures to support the conclusions obtained by the change of the pore structure of PDMS.

2. In the manuscript, the author should give a detailed description of the basic principles of foaming to form a porous structure, not just a description of the process

3. The fabricated PDMS foam has low density and good mechanical properties. The authors are suggested to compare the previously reported works about the PDMS foam in the revised manuscript.

4. The authors focus on the influence of m-HM contents on the performance of PDMS foam, how about the size of the m-HM?

5. Typo in line 161 of page 4 : " 5-100 μn" should be "5-100 μm ". 

Author Response

Revisions made in light of comments made by Reviewer #1:

Reviewer #1: I suggest publication after revision. My comments are below:

Authors Reply: Many thanks for your professional review work on our manuscript that helps us to improve the paper substantially. According to your valuable suggestion, we have made careful corrections all throughout the paper. The detailed point-by-point responses as shown in blue color fonts are listed below.

Specific comments:

Comment 1. In the manuscript, with the increase of m-Hm content, there is not enough data and pictures to support the conclusions obtained by the change of the pore structure of PDMS.

Reply 1: Thanks for your advice. During the synthesis of SiRF@m-HM composites, the amounts of silylation reagent are fixed at 5 wt%, while m-HM contents are in the range of 2.5 wt%-30 wt%. As shown in Figure 4, different sample morphologies can be obtained by introducing the m-HM content. Digital photos clearly indicate that a strong relationship between foam density and the amount of m-HM exists. In other word, when the m-HM loading is less than 7.5 wt%, the higher the m-HM content produces the lower density of the PDMS foam composites. Moreover, the foam pore size and foaming rate (final foam volume: initial mixed solution volume) increase in line with increased m-HM content (Figure 4a-4f), and the foam network structure is damaged gradually. It is hypothesized that a higher amount of m-HM in the PDMS phase could inhibit the curing reaction rate between Si-H of PDMS-H and Si-vi of PDMS-Ph-vi [1]. As the previous studies, the most significant factor was the balance of reaction rates of foaming and cross-linking [2, 3]. Due to the slower curing process, hydrogen gas from the dehydrogenation reaction between Si-OH of PDMS-Ph-OH or H2O and Si-H of PDMS-H converges into more giant bubbles or escapes directly into the air, which leads subsequently to obtain the SiRF foam composites with an irregular network structure and a relatively high density (Figures 4e and 4f).

Comment 2. In the manuscript, the author should give a detailed description of the basic principles of foaming to form a porous structure, not just a description of the process.

Reply 2: Thanks for your advice. We have systematically expounded the principles of foam to form a porous structure in our previous research, e.g. DOI: 10.1016/j.cej.2020.124724 and DOI: 10.1039/c9ta09372a. Therefore, we would no describe it again in the manuscript.

Comment 3. The fabricated PDMS foam has low density and good mechanical properties. The authors are suggested to compare the previously reported works about the PDMS foam in the revised manuscript.

Reply 3: Good suggestion. According to the Reviewer’s suggestion, we have added the related content of the Table 1 in the revised manuscript as following:

Table 1. Comparisons of processing condition, density, mechanical properties (strainmax and stress) and thermally insulating properties (T1/T2/thickness).

Type a

Processing condition

Density (g/cm-3)

Strainmax (%)/

Stress (kPa)

T1 (°C)/T2 (°C)/

thickness (mm) c

Ref.

Graphene/PDMS

Freeze-drying,

annealing at 200 °C

74

~90%/~1845

61.2/200/-

[4]

ST-8-0.6

3D printing,

drying at 80 °C

320

~60%/~400

NM

[5]

S46-40

Mechanical mixing,

curing at 200 °C

~800

-/~11000

NM

[6]

PDMS-40%/RFSi-0.6

“Co-gel” technique,

drying at 80 °C

NM b

20%/699

~175/-/4

[7]

BNF@PDMS

Ni template,

curing at R.T.

15

~80/~1.5

NM

[8]

Sample-11

Mechanical mixing,

foaming at 70 °C

346

~70%/~520

NM

[9]

SiRF-m-7.5

Mechanical mixing, foaming at R.T.

104

80%/58

52.1/150/20

This work

a ST: simple tetragonal; S46-40: 40 wt% hollow glass microballoons of S46; RF: resorcinol formaldehyde; h-BNFs: hexagonal boron nitride foams; Sample-11: PDMS base/Span 80.

b NM: Not Mentioned.

c T1: top temperature of objective; T2: test temperature.

Comment 4. The authors focus on the influence of m-HM contents on the performance of PDMS foam, how about the size of the m-HM?

Reply 4: Thanks for your careful check. The hollow microsphere described in Figure 2, and “The commercial HM particles” have been revised to “The modified HM particles” in the manuscript. We would not show any information on the commercial HM particles in the manuscript due to no difference between SEM image of the modified HM particles and image of the commercial HM particles (Figure S1).

Figure S1. SEM image of the commercial HM particles

Comment 5. Typo in line 161 of page 4: " 5-100 μn" should be "5-100 μm ". 

Reply 5: Thanks for your careful check. According to the Reviewer’s suggestion, we have marked in the Section 3 of results and discussion.

Reviewer 2 Report

see attached file

Author Response

Reviewer #2: The authors analyze “the use of hollow microparticles as an effective strategy for fabricating lightweight, mechanically flexible and thermal insulation PDMS foam composite materials for many potential applications.” They describe the basic ideas and methods how this approach can be realized and the advantages of this method. The paper is well-motivated and well-performed. The presentation is, in general, fine as well, however, including some minor bugs like the one’s listed below:

* Correspondence: authors (9)

- foam mateiral (49)

- (v/v=1/9) (111)

- 650 °C. (136)

- and present a high-viscous feature (195)

- Structure and morphlogy (232)

I recommend acceptance after minor correction. A second review is, from my point of view, not required.

Author Reply: We would like to thank the reviewer for his/her positive comments that can help us greatly improve the quality of this manuscript. A point-by-point response to each comment is given below, and revisions in the manuscript are indicated by blue color fonts.

Specific comments:

Reply 1: Thanks for your careful check. According to the Reviewer’s suggestion, we have marked in new manuscript.

Reviewer 3 Report

The manuscript titled "Facile synthesis of hollow glass microsphere filled PDMS foam composites with exceptional lightweight, mechanical flexibility, and thermal insulating property" is a good idea and wall presented article by Tian-Long Han and coworkers. I could not find any point of low merit for the rejection of this paper on publication. I have only two minor points to be addressed in this article before its publication, as given below. Overall this paper is well to publish in Journal like molecules ( if published in special issue related to polymers).

The Ist point of concern is the use of abbreviations in the title of the paper.

second concern is, Authors need to provide some literature comparison of thermal and mechanical properties (In a table) with their results to prove the novelty.

Author Response

Reviewer #3:

The manuscript titled "Facile synthesis of hollow glass microsphere filled PDMS foam composites with exceptional lightweight, mechanical flexibility, and thermal insulating property" is a good idea and wall presented article by Tian-Long Han and coworkers. I could not find any point of low merit for the rejection of this paper on publication. I have only two minor points to be addressed in this article before its publication, as given below. Overall this paper is well to publish in Journal like molecules (if published in special issue related to polymers).

Author Reply: Thank you very much for your kind proposals. According to your valuable advice, we have made careful corrections all throughout the paper. The detailed point-by-point responses as shown in blue color fonts are listed below. We have addressed the point-by-point responses.

Specific comments:

Comment 1. The 1st point of concern is the use of abbreviations in the title of the paper.

Reply 1: Thanks for your advice. According to the Reviewer’s suggestion, we have carefully checked the title.

Comment 2. The second concern is, Authors need to provide some literature comparison of thermal and mechanical properties (In a table) with their results to prove the novelty.

Reply 2: Good suggestion. According to the Reviewer’s suggestion, we have added the related content of the Table 1 in the revised manuscript as following:

Based on the result, we compare the density, processing condition, mechanical properties, and thermally insulating properties of various PDMS foam systems, which are given in Table 1. It can be seen that our foam not only a simple fabrication possesses and mechanical flexibility, but also exhibits ultra-low density and excellent thermally insulating properties.

Table 1. Comparisons of processing condition, density, mechanical properties (emax and stress) and thermally insulating properties (T1/T2/thickness).

Type a

Processing condition

Density (g/cm-3)

emax (%)/

Stress (kPa) c

T1 (°C)/T2 (°C)/

thickness (mm) d

Ref.

Graphene/PDMS

Freeze-drying,

annealing at 200 °C

74

~90%/~1845

61.2/200/-

[4]

ST-8-0.6

3D printing,

drying at 80 °C

320

~60%/~400

NM

[5]

S46-40

Mechanical mixing,

curing at 200 °C

~800

-/~11000

NM

[6]

PDMS-40%/RFSi-0.6

“Co-gel” technique,

drying at 80 °C

NM b

20%/699

~175/-/4

[7]

BNF@PDMS

Ni template,

curing at R.T.

15

~80/~1.5

NM

[8]

Sample-11

Mechanical mixing,

foaming at 70 °C

346

~70%/~520

NM

[9]

SiRF-m-7.5

Mechanical mixing, foaming at R.T.

104

80%/58

52.1/150/20

This work

a ST: simple tetragonal; S46-40: 40 wt% hollow glass microballoons of S46; RF: resorcinol formaldehyde; h-BNFs: hexagonal boron nitride foams; Sample-11: PDMS base/Span 80.

b NM: Not Mentioned.

c emax: Maximum strain

d T1: top temperature of objective; T2: test temperature.
